# The Impact of Parenting Avoidance (IPA): Scale Development and Psychometric Evaluation Among Parents of Transgender Youth

**DOI:** 10.3390/bs15050625

**Published:** 2025-05-03

**Authors:** Haley R. Hedrick, Stephanie V. Caldas, Danielle N. Moyer

**Affiliations:** 1Department of Pediatrics, Oregon Health & Science University, 707 SW Gaines Street, Portland, OR 97239, USA; hhedrick19@georgefox.edu; 2Department of Clinical Psychology, George Fox University, Newberg, OR 97132, USA; 3Department of Child and Adolescent Psychiatry, Hassenfelds Children’s Hospital at NYU Langone, New York, NY 10016, USA; stephanie.caldas@nyulangone.org

**Keywords:** psychological flexibility, parenting, avoidance, parents, youth, transgender, adolescents, parental support

## Abstract

Parental support and acceptance are strong protective factors for better mental health outcomes among transgender and gender diverse youth. Psychological inflexibility, specifically in the role of parenting, or “parenting inflexibility”, refers to an over-reliance on avoidance strategies at the expense of parenting values. Parenting inflexibility may be related to parental support, making it a useful target of intervention for parents of transgender youth. The aim of the present study was to develop a brief clinically useful measure of parenting inflexibility based on a synthesis of existing measures and to evaluate the psychometric properties across two study populations. Study 1 used exploratory factor analysis to examine this measure among parents in the general population recruited using MTurk. Study 2 used confirmatory factor analysis to examine the measure among parents of transgender youth recruited from a clinic. The final measure, the Impact of Parenting Avoidance (IPA) scale, is a one-factor 7-item measure of parenting inflexibility that is easy to administer and interpret in a pediatric health setting. The resulting measure demonstrated acceptable reliability, and it was significantly correlated with important outcome variables, such as negative parenting practices and lower perceived parental support among transgender and gender diverse youth.

## 1. Introduction

Health disparities and adverse health outcomes among transgender and gender diverse (TGD) youth result from intersectional psychosocial risk factors, including discrimination and minority stress ([19]), deficits in mainstream education on gender diversity ([34]), a high prevalence of childhood trauma ([20]), and lack of access to competent comprehensive gender-affirming healthcare ([25]; [47]). For some TGD youth, access to early multidisciplinary gender-affirming healthcare significantly improves psychosocial wellbeing, including decreased suicidal ideation and increased quality of life and peer relations ([18]; [57]). However, TGD youth presenting to pediatric gender clinics often do so with elevated rates of anxiety, depression, and suicidality ([41]), and simply establishing care alone, without subsequent interventions, does not alleviate psychological distress ([8]).

Strong evidence shows that gender-affirming treatments lead to significant reductions in distress and improved wellbeing, largely related to improvements in body satisfaction ([10]; [35]) and associated strengthening of neural connections that facilitate emotion regulation ([23]). While decades of research demonstrate the benefits of gender-affirming care for most youth, it is unclear which factors contribute to the smaller subset of youth who do not demonstrate improvements. Emerging research suggests that parental support and acceptance may be an important moderating factor in this relationship ([11]), which is consistent with broader research establishing parental support as a strong protective factor against a variety of negative outcomes ([7]; [44]; [50]). For the purposes of this paper, the terms parent and parenting will be used to encompass all parent, caregiver, and guardian roles related to child-rearing.

In a large, longitudinal study of pediatric gender-affirming care, [11] ([11]) found that among youth with more psychological distress prior to gender-affirming interventions, those with adequate parental support were significantly more likely to show improvements over the course of treatment. Conversely, TGD youth with more distress at baseline and low parental support did not show similar improvements over time. Interventions that improve parental support and acceptance may therefore be important concurrent targets for clinicians working within pediatric gender clinics. By identifying intervenable factors related to parent support and acceptance that are easily assessed within a multidisciplinary clinic setting, interventions can be appropriately adapted specifically for parents of TGD youth accessing gender-affirming care.

Psychological flexibility (and conversely inflexibility) is a transdiagnostic behavioral process that is relevant to many areas of psychological distress, wellbeing, performance, and functioning, and it is the primary target of third-wave behavioral interventions such as acceptance and commitment therapy (ACT; [28]; [36]). Psychological flexibility involves acting mindfully and consistently with personal values, even in the presence of psychological distress or discomfort, while psychological inflexibility involves an over-reliance on avoidance coping at the expense of values-based actions ([13]). The term experiential avoidance has sometimes been used interchangeably with psychological inflexibility ([28]) as it is a major component of this coping style, but there are other aspects to inflexibility that are not fully captured by this term. Specifically, there is a behavioral aspect wherein a person acts inconsistently with their values (or does not act at all) as a result of over-reliance on avoidance. Psychological flexibility has been studied in a variety of parenting populations ([6]) as well as among TGD adults ([30]), but it has not been examined among parents of TGD youth.

Psychological flexibility, specifically in the context of parenting, or parenting flexibility, is the ability to act consistently with parenting values in the presence of discomfort related to parenting, while parenting inflexibility is characterized by avoidance and self-judgment related to parenting ([6]). As with general inflexibility, avoidance is a major component of parenting inflexibility, but there are other aspects to inflexibility and, in particular, a behavioral impact of the over-reliance on avoidance. Parents who are more psychologically flexible demonstrate more adaptive parenting, have lower parenting stress and greater family cohesion, and have children who demonstrate lower psychological distress ([17]). Interventions aimed at improving parenting flexibility have resulted in positive outcomes for both parents and children ([5]). Interventions such as ACT that target parenting flexibility may focus on helping parents clarify their parenting-related values, practice mindfulness in the face of challenging family interactions, and notice when self-judgments or rigid societal standards are influencing parenting practices instead of personally held values.

Research has shown that emotional barriers, as well as personal beliefs and attitudes, are common barriers to support and acceptance among parents of TGD youth ([38]), suggesting that willingness to tolerate distress and clarification of parenting-specific values may facilitate improvements in parental support. Mindfulness and acceptance-based interventions such as ACT may therefore be particularly beneficial for some parents of TGD youth seeking gender-affirming care. Efficiently identifying parenting flexibility, and in particular parenting inflexibility, will help to allocate resources to the parents who would most benefit from these interventions. To this end, a brief measure of parenting flexibility that can be easily administered in a multidisciplinary clinic setting in which multiple goals for intervention are being identified would be clinically useful.

Available measures of parenting flexibility include the Parental Acceptance and Action Questionnaire (PAAQ; [12]), the Parental Psychological Flexibility (PPF) Questionnaire ([4]), the Parental Acceptance Questionnaire (6-PAQ; [24]), and the Parenting-Specific Psychological Flexibility (PSPF) scale ([2]). There are also measures that focus on more specific contexts, such as parenting a child with chronic pain (e.g., [58]), as well as translations of the available measures in select languages. Other than these more specific measures, there is limited evidence to support which measure of parenting flexibility is most appropriate based on population, assessment objectives, or context of administration ([6]).

While these available measures have all demonstrated some degree of reliability and validity, they vary in regard to development methods, length, answer choice options, aspects of psychological flexibility, and the inclusion of negatively worded items, among others. Some items appear across multiple available measures, while many others differ, and no measure was developed based on accessibility or reading level. Some measures contain items most appropriate to a specific age range, and some contain items that may be conflated with general psychological distress, a common criticism of all measures of psychological flexibility ([13]).

In the context of multidisciplinary pediatric gender-affirming care specifically, identifying parenting behaviors or attitudes that may facilitate or hinder support and acceptance of a TGD child is clinically useful to the clinical team. Parenting inflexibility may be a particularly useful factor to identify because there are well-established interventions that can be implemented to improve flexibility. ACT has been shown to be effective, even with brief versions of the intervention, with parents ([27]) and in healthcare settings specifically (e.g., [43]). Other pediatric medical teams that include integrated behavioral health may similarly benefit from identifying parenting behaviors that have the potential to impact care. Clinicians in these settings would benefit from a brief instrument to identify clinically relevant parenting inflexibility efficiently across child and adolescent developmental levels.

The goals of the present study were therefore to (a) examine existing measures of parenting flexibility to identify strengths, weaknesses, and overlapping features, (b) synthesize this information for the purpose of designing a broadly applicable and clinically useful measure of parenting inflexibility, and (c) examine the relationship between parenting flexibility and parental support and acceptance among parents of TGD youth. We aimed to develop a measure that is quick to administer, easy to understand, and efficient with regard to interpretation and responding to results.

## 2. Materials and Methods

This study consisted of three phases: (1) the synthesis of prior parenting flexibility scales and construction of a new measurement tool, (2) initial validation and exploratory factor analysis (EFA) of the measure among a non-clinical sample of parents recruited through Amazon Mechanical Turk (MTurk), and (3) further validation and confirmatory factor analysis (CFA) of the measure among a clinical population of parents presenting with their child for multidisciplinary gender-affirming care.

### 2.1. The Impact of Parenting Avoidance (IPA) Scale Development

Scale development followed established design methods ([21]), including (1) articulation of construct and context, (2) choosing response format and assembling an initial item pool, (3) collection of data, and (4) examination of psychometric properties and quality. In addition to articulating the construct and context (step 1), we conducted a scoping review of existing measures of parenting flexibility, the results of which can be found in [6] ([6]). To facilitate the development of the response format and item pool (step 2), we synthesized the existing measures of parenting flexibility to identify overlapping content and useful discrepancies, synthesize the information into a new pool, and further refine the construct definition and focus. Data collection (step 3) and examination of psychometric properties (step 4) were completed across two studies with different parent populations.

#### 2.1.1. Initial Articulation of Construct and Context

While the primary goal of this project was to develop a measure of psychological flexibility for parents of TGD youth seeking gender-affirming care, we decided to define the context more broadly as integrated pediatric medical settings. This definition of the context is appropriate for the target audience while facilitating a more broadly applicable measure, capturing a context that has not previously been a focus of similar measures. We initially defined the target construct in terms of both parenting flexibility and inflexibility, leaving versatility for the final measure based on the results of prior measurement synthesis and overall refinement of the construct definition over the course of the measurement development process.

Parenting flexibility was initially defined as a broad pattern of behavior characterized by effective management of psychological experiences related to parenting in the service of acting consistently with parenting-specific values. Parenting inflexibility was similarly defined as a broad pattern of behavior characterized by the tendency of psychological experiences related to parenting to interfere with acting consistently with parenting-specific values. Existing English-language measures that matched this initial construct definition were identified for examination, synthesis, and potential use in the new measure, which included the PAAQ ([12]), the PPF Questionnaire ([4]), the 6-PAQ ([24]), and the PSPF scale ([2]). See Table 1 for a summary of these measures, including number of items, response format, directionality of the total score (flexibility versus inflexibility), subscales, and inclusion of reverse scoring. More information about each measure, including target population, validation results, strengths, and limitations, can be found in [45] ([45]) and [6] ([6]).

#### 2.1.2. Generation of the Item Pool and Response Format

Items from the four existing measures of parenting flexibility and the most widely used measure of general psychological flexibility, the acceptance and action questionnaire, second edition (AAQ-II; [1]), were reviewed and compared for overlap and deviation and examined thematically. Items from the AAQ-II were only used to compare themes, as no items of general flexibility were intended to be included in the new measure. Items were systematically included in the thematic analysis if they occurred across several measures or occurred on only one measure but clearly fit the broad definition of parenting flexibility defined above.

Items were excluded from thematic analysis for the following reasons: (1) content related to parenting broadly but not specifically to flexibility, for example, “I am consistent in my parenting practices”, (2) content related to distress broadly but not specifically to flexibility, for example, “I get upset if things don’t go my way when I interact with my child”, (3) content confined to a particular age group, for example, “I avoid taking my child to the store for fear of how they will behave”, (4) item contained a double negative or other potentially confusing content, for example, “I’m not afraid of my child’s feelings”, and (5) content not clearly related to parenting flexibility for some other reason, for example, “Worries get in the way of my child’s success”. The remaining items were reviewed thematically to identify patterns in item content and relationship to overall content. One overarching theme that emerged was that several items contained two components: (1) an aspect of avoidance and (2) the impact of that avoidance. For example, both the PSPF scale and the PPF questionnaire contain the item “[My] worries get in the way of my success as a parent”.

Items were then categorized according to these two components. Avoidance component themes included general painful experiences (e.g., “My painful memories prevent me from having a fulfilling life as a parent”), negative self-evaluations related to parenting (e.g., “I often catch myself daydreaming about things I’ve done with my child and what I would do differently next time”), worries related to the child (e.g., “In order for my child to do something important, I have to have all my doubts about it worked out”), and experiences with the child’s own distress (e.g., “I try hard to avoid having my child feel depressed or anxious”). Impact component themes included impacts to the parent (e.g., “I could not cope with the guilt if my child did something wrong”), impacts to the child (e.g., “I don’t let my child do things that I’ll worry about”), and impacts to parenting ability (e.g., “My feelings stop me from doing what is best for my children”). Some items were excluded at this stage due to the absence of an impact component, for example (e.g., “It seems to me that most people are better parents than I am”).

Based on the results of thematic analysis, the goal of identifying parents who may need intervention, and considerations for readability and ease of use, it was decided at this stage that the measure would aim to capture parenting *inflexibility*. The themes of avoidance and impact also reflected our initial definition of parenting inflexibility, which was a broad pattern of behavior characterized by the tendency of psychological experiences related to parenting to interfere with acting consistently with parenting-specific values. Therefore, no items reflecting flexibility or requiring reverse coding were included. An initial item pool was developed in an attempt to reflect each of the themes listed above. Finally, the results of this thematic analysis of existing measures also informed the new scale name: the Impact of Parenting Avoidance (IPA) scale.

A total of 10 items were constructed to match each of the themes listed above, by choosing the language used in an existing measure, combining items from multiple existing measures, or taking a relevant component from an existing measure and revising the item to better reflect the two components of avoidance and impact. These ten items were then further revised to improve readability, aiming for a maximum reading level of eighth grade ([32]). Following this round of revisions, items ranged from grade 3.8 to grade 7.2 on the Flesch-Kincaid (F-K) grade level scale, and the full 10 items together resulted in an F-K grade level of 5.5.

Response format choices were based on the Patient-Reported Outcomes Measurement Information System (PROMIS; [9]), which is used widely across various health domains due to the ease of use, standardization, and consistent scoring metric. This response format consists of the following five-point scale: (1) Never, (2) Rarely, (3) Sometimes, (4) Usually, and (5) Always. By formatting responses consistently with other brief measures that parents may be completing in the same clinic or other clinics across their healthcare system, we hoped to further decrease the potential for misunderstanding of instructions and improve the accuracy of the results.

This initial 10-item IPA scale with a five-point response format was sent to two professionals with expertise in the construct of psychological flexibility, specifically among children and parents, for review and feedback. These two experts were asked to rate each item on how closely it represented the construct (1—minimal, 2—moderate, 3—strong) and overall clarity (1—poor, 2—moderate, 3—clear) and to provide overall feedback and recommendations. Based on this feedback, one item was removed and several additional items were revised. See Table 2 for a complete list of the initial 9-item measure with F-K reading level statistics. The reading level of one item did increase after revision (see Table 2), but the full measure remained at a fifth-grade reading level, which was deemed appropriate.

### 2.2. Study 1 Methods: Exploratory Factor Analysis with the MTurk Sample

The first psychometric evaluation of the IPA scale utilized MTurk to recruit a broad non-clinical sample of parents in the United States. This was a cross-sectional study that explored the initial factor structure of the IPA, followed by an examination of the reliability and validity of the resulting measure. All procedures, including informed consent, were approved by an institutional review board.

#### 2.2.1. Participants and Procedures

MTurk current-generation UI was used to recruit parents with at least one child between the ages of 11 and 18 years old. MTurk is a crowdsourcing software that has demonstrated utility in recruitment for behavioral science research, including clinical and subclinical samples ([53]). Specifically, high-quality data can be collected quickly at a relatively low cost across a diverse group of individuals based on study-specific parameters and variables of interest. MTurk has been used in similar prior research on parenting flexibility due to these advantages, as well as parent-specific advantages such as the ability to adequately survey fathers in addition to mothers ([2]; [46]). We took several additional steps to help ensure the quality of data collected through MTurk, including providing adequate compensation and multiple attention checks throughout the survey and employing a high degree of scrutiny for screening out respondents who were not likely to have answered the survey truthfully. Based on a likely one-factor structure with high communalities, we collected responses until at least 100 participants were deemed appropriate for inclusion in the study ([22]).

#### 2.2.2. Additional Measures

In addition to the IPA scale, several additional measures were completed by participants via MTurk. This included the four existing measures of parenting flexibility described above, the AAQ-II, which is the most commonly used measure of general psychological flexibility, as well as additional measures of parenting behaviors, distress, and social support.

*Existing measures of parenting flexibility*. The four existing measures of parenting flexibility, on which the IPA was designed (PAAQ, PPF, 6-PAQ, and PSPF), were included. The length, response format, and scoring information for each of these measures can be found in Table 1. Each of these measures has demonstrated validity and reliability in prior samples ([6]). In the current sample, internal consistency reliability was questionable for the full-scale PAAQ (α = 0.64) and the Inaction subscale (α = 0.65), and it was quite poor for the Unwillingness subscale (α = 0.334). Internal consistency was good for the full scale 6-PAQ (α = 0.88), as well as for the subscales of Acceptance (α = 0.83) and Defusion (α = 0.84), acceptable for Self-As-Context (α = 0.74), and questionable for Values (α = 0.68) and Committed Action (α = 0.60). Internal consistency was poor for Being Present (α = 0.43). Only the full-scale scores for the PAAQ and 6-PAQ were used for the remaining analyses. For the PPF, internal consistency was excellent for the Cognitive Defusion subscale (α = 0.96) and good for the Committed Action subscale (α = 0.89) and the Acceptance subscale (α = 0.86). As the PPF does not yield a singular total score, the three subscales were used in the remainder of the analyses.

During data analysis, it was discovered that one item of the PSPF scale (“In my role as a parent, I worry about not being able to control my worries and feeling”) was omitted from the survey, and therefore, a 6-item version of the measure was used in the analysis. While the developers of the PSPF suggest that all items can be reverse-coded to reflect psychological flexibility (see Table 1), original coding was used in this study, as there was no theoretical utility in reversing the scoring for the purposes of this study. Internal consistency reliability for the 6-item PSPF scale of psychological inflexibility was excellent (α = 0.96).

*General psychological inflexibility*. The AAQ-II ([1]) is a 7-item measure of general psychological inflexibility, also sometimes referred to as experiential avoidance. Respondents are asked how true each item is for them on a 7-point scale from 1 (never true) to 7 (always true). An example item is “My painful experiences and memories make it difficult for me to live a life that I would value”. Items are summed to create one total score, with higher scores indicating higher inflexibility. The AAQ-II has demonstrated validity and reliability across diverse samples ([1]; [15]). In the current sample, internal consistency reliability was excellent for the AAQ-II (α = 0.96).

*Parenting behaviors*. The Multidimensional Assessment of Parenting Scale (MAPS; [46]) is a 34-item measure of positive and negative parenting practices. Respondents are asked to rate each item for how well it describes their parenting practices over the past two months from 1 (never) to 5 (always). Average scores across 16 items make up Broadband Positive Parenting, which can be further divided into four subscales. Example items for each subscale include “I avoid struggles with my child by giving clear choices” (Proactive Parenting), “If my child cleans his room, I will tell him/her how proud I am” (Positive Reinforcement), “My child and I hug and/or kiss each other” (Warmth), and “I listen to my child’s ideas and opinions” (Supportiveness). Average scores across the remaining 18 items make up Broadband Negative Parenting, which can be further divided into three subscales. Example items for each subscale include “I argue with my child” (Hostility), “I feel that getting my child to obey is more trouble than it’s worth” (Lax Control), and “I spank my child when I am extremely angry” (Physical Control). The MAPS has demonstrated strong psychometric properties, including parents who identify as LGBTQIA+ ([46]; [51]). In the current sample, internal consistency reliability was good for Broadband Positive Parenting (α = 0.88) and excellent for Broadband Negative Parenting (α = 0.97).

*Depression, anxiety, and stress*. The 21-item version of the Depression Anxiety and Stress Scales (DASS; [37]) was used to assess psychosocial distress. Respondents are asked how much each item applied to them over the past week on a 4-point scale from 0 (did not apply to me at all) to 3 (applied to me very much or most of the time). Total scores are calculated by summing 7 items for each subscale. Example items for each subscale include “I felt that I had nothing to look forward to” (Depression), “I was aware of dryness of my mouth” (Anxiety), and “I found it hard to wind down” (Stress). The DASS-21 has shown strong psychometric properties across diverse samples, including parents (e.g., [42]). In the current sample, internal consistency reliability was excellent for all three scales of Depression (α = 0.96), Anxiety (α = 0.95), and Stress (α = 0.94).

*Social support.* The 4-item PROMIS Emotional Support Short-Form ([26]) was used to measure social support. Respondents are asked to respond to each item on a scale from 1 (never) to 5 (always). Items are summed to create a total score, with higher scores indicating more support. An example item is “I have someone to talk with when I have a bad day”. In the current sample, internal consistency reliability was acceptable for social support (α = 0.75).

#### 2.2.3. Analysis Plan

A factor analysis was conducted using generalized least squares extraction and no rotation ([22]). Mean substitution was used for missing data, as it has been deemed better than deletion methods ([39]). Examination of the scree plot and coefficient matrix was used to evaluate factor structure. Following the EFA, validity of the resulting item pool was examined with correlation analyses comparing the IPA to existing measures of parenting inflexibility (convergent validity), parenting practices and distress (concurrent validity), and social support (discriminant validity). Regression analysis was used to determine whether the IPA predicted parenting practices or distress over and above general psychological inflexibility (incremental validity). Reliability was examined using internal consistency. Additional regression analyses were also run to clarify whether any one of the measures of parenting flexibility was superior at predicting distress or parenting practices.

### 2.3. Study 2 Methods: Confirmatory Factor Analysis with the Clinical Sample

The second psychometric evaluation of the IPA scale utilized clinical data from parents of TGD youth attending a multidisciplinary pediatric gender clinic. This was a cross-sectional study of the final factor structure, reliability, and validity of the IPA. All procedures were approved by an institutional review board. Only a retrospective chart review was used to extract data, and no direct participant involvement occurred. Informed consent was therefore deemed not applicable to Study 2 by the review board.

#### 2.3.1. Participants and Procedures

Adolescents seeking care in our pediatric gender clinic and/or their parents (depending on age and other factors) regularly complete screening measures as part of routine appointments with the team psychologist. A retrospective chart review of these clinical measures was later conducted for patients who attended appointments between April 2020 and July 2023. As the primary focus of this study was the development of a measure for parents, cases with youth questionnaires only or incomplete parent-reported IPA data were excluded. Cases with parent questionnaires only were included for use in the CFA, but a smaller sample of cases with matched parent and adolescent data was used for additional analyses. One additional case was removed for being an outlier on the total IPA score.

#### 2.3.2. Additional Measures

In addition to the IPA scale, several additional measures are completed by patients and parents when they have an appointment with the gender clinic psychologist. For the purposes of this study, adolescent-reported depression, anxiety, and perceived parental support were also included in the analysis.

*Adolescent depression*. The Patient Health Questionnaire (PHQ-9; [54]) is a 9-item measure of depressive symptoms that has been modified for appropriateness among adolescents (PHQ-A). Respondents are asked how often they have been bothered by each symptom over the past two weeks on a 4-point scale from 0 (Not at all) to 3 (Nearly every day). An example item is “Feeling down, depressed, or hopeless”. Items are summed for a total depression score. The PHQ-9 has demonstrated strong reliability and validity in other samples ([49]), and it has been shown to be clinically useful in transgender youth presenting to pediatric gender clinics ([41]). Internal consistency reliability in the current sample was excellent (α = 0.90).

*Adolescent anxiety*. The Generalized Anxiety Disorder version of the Patient Health Questionnaire (GAD-7; [55]) is a 7-item measure of anxiety symptoms. Respondents are asked how often they have been bothered by each symptom over the past two weeks on a 4-point scale from 0 (Not at all) to 3 (Nearly every day). An example item is “Feeling nervous, anxious, or on edge”. Items are summed for a total depression score. The GAD-7 has demonstrated strong reliability and validity in other samples ([40]) and it has been shown to be clinically useful in transgender youth presenting to pediatric gender clinics ([41]). Internal consistency reliability in the current sample was excellent (α = 0.90).

*Perceived parental support*. The Perceived Parental Attitudes of Gender Expansiveness (PAGES-Y; [29]) is a 14-item measure of perceived parental support. Respondents are asked how much they agree with each item on a 5-point scale from 1 (Strongly Disagree) to 5 (Strongly Agree). The measure produces two subscales. Parental Non-Affirmation is calculated by summing 8 items, an example of which is “My parent/caregiver has problems with my gender expression”. Parental Acceptance is calculated by summing the remaining 6 items, an example of which is “My parent/caregiver is supportive of my gender transition”. The PAGES-Y has demonstrated reliability and validity. Internal consistency reliability in the current sample was excellent for Non-Affirmation (α = 0.92) and good for Acceptance (α = 0.88).

#### 2.3.3. Analysis Plan

Confirmatory factor analysis of the 8-item IPA was completed using RStudio with R Statistical Software (version 4.0.5; [48]) with the Lavaan (version 0.6-19) package ([52]) using maximum likelihood estimation, and the following fit indices were used: chi-square (c^2^), comparative fit index (CFI; general cutoff greater than or equal to 0.90), Tucker–Lewis Index (TLI; general cutoff greater than or equal to 0.90), root mean square error of approximation (RMSEA; general cutoff less than or equal to 0.06), and Bartlett’s test of sphericity (Bartlett’s; general cutoff less than or equal to 0.05; [56]). Fit indices were interpreted intentionally with caution, not to treat all cutoff values as completely objective standards ([31]). Following the CFA, validity was examined with correlation analyses comparing the IPA to adolescent-reported depression, anxiety, and perceived parental support. Reliability was examined using internal consistency. A post-hoc CFA of the Study 1 data was run to evaluate the hypothesis that one poor-fitting item may have had population-specific implications.

## 3. Results

### 3.1. Study 1 Results: Exploratory Factor Analysis with the MTurk Sample

A total of 101 adults who have at least one child between the ages of 11 and 18 completed the online surveys. Data from an additional 33 individuals were excluded from the current sample because they did not meet adequate validity standards. Participants represented a heterogeneous sample with regard to age, gender, and racial/ethnic identity, as expected with an MTurk sample. In contrast, the sample was over-represented by adults who were married or living with a partner and those with higher educational attainment. Education level was included as a proxy for socioeconomic status and therefore suggests the sample was skewed toward higher SES. See Table 3 for a full summary of participant characteristics.

#### 3.1.1. Exploratory Factor Analysis

No outliers were identified in total IPA scores according to standardized residuals (i.e., none greater than 3.3), and all additional assumptions for factor analysis were met. There were 4 participants with one missing value, for whom a mean substitution method was used ([39]). A factor analysis was conducted using generalized least squares extraction and no rotation. Examination of the scree plot and coefficient matrix supported a one-factor construct, consistent with the hypothesis. One item (“I try hard to avoid having my child feel sad or anxious”.) was identified as fitting poorly with the factor and was subsequently removed.

#### 3.1.2. Reliability and Validity

Internal consistency of the resulting 8-item IPA was excellent (α = 0.91). Correlation analysis found strong convergent validity with measures of psychological inflexibility, including the PAAQ (*r* = 0.79, *p* < 0.001), 6-PAQ (*r* = 0.82, *p* < 0.001), and PSPF (*r* = 0.90, *p* < 0.001). The IPA was also negatively correlated with psychological flexibility on the Cognitive Defusion (*r* = −0.91, *p* < 0.001) and Committed Action (*r* = −0.88, *p* < 0.001) subscales of the PPF but not the Acceptance subscale (*r* = −0.04, *p* = 0.681). Contrary to our hypothesis, the IPA *was* correlated with social support (*r* = 0.69, *p* < 0.001), and we were unable to determine discriminant validity in this study. Social support was also correlated in unexpected directions with other measures of distress (DASS *r* = 0.71, *p* < 0.001) and psychological flexibility (AAQ *r* = 0.76, *p* < 0.001; PAAQ *r* = 0.587, *p* < 0.001; PPFQ *r* = −0.62, *p* < 0.001; PAQ *r* = 0.58, *p* < 0.001; PFPS *r* = 0.68, *p* < 0.001).

Regarding concurrent validity, the IPA was positively correlated with depression (*r* = 0.82, *p* < 0.001), anxiety (*r* = 0.82, *p* < 0.001), stress (*r* = 0.80, *p* < 0.001), and negative parenting practices (*r* = 0.85, *p* < 0.001), but there was no significant correlation between parenting inflexibility and positive parenting practices (*r* = −0.15, *p* = 0.148). Linear regression was used to evaluate incremental validity. P-P plots and residual scatterplots were used to examine normality and homoscedastic, respectively. To address multicollinearity between DASS subscales, they were combined into a singular distress variable when used as an independent variable but kept as separate anxiety, depression, and stress scales when examined as dependent variables. The IPA demonstrated incremental validity over and above general psychological flexibility when predicting negative parenting practices (β = 0.48, *R*Δ = 0.04, *p* < 0.001), but it did not add incremental validity when predicting depression, anxiety, or stress. The IPA continued to predict negative parenting practices when controlling for general psychological flexibility and distress (β = 0.47, *R*Δ = 0.03, *p* = 0.002). See Appendix A for additional details on the full analyses.

#### 3.1.3. Comparison of All Measures of Parenting Flexibility

Multiple linear regression models were conducted to examine whether other measures of parenting flexibility would demonstrate incremental validity and whether one would demonstrate superiority to others. P-P plots and residual scatterplots were examined for normality and homoscedasticity, respectively. Some degree of multicollinearity was expected given the overlap in measurement tools. DASS subscales were combined into a singular distress variable when used as an independent variable to reduce multicollinearity, given there was no theoretical rationale to examine them separately, whereas parenting inflexibility measures were meant to be examined alongside one another. When multicollinearity remained high (VIF > 10), analyses were excluded or interpreted with caution.

No measures predicted Stress over and beyond general psychological flexibility. The PSPF, 6-PAQ, and the Cognitive Defusion subscale of the PPF predicted anxiety and depression. However, in a model where these measures were controlling for one another, only the PSPF remained significant in predicting anxiety (β = 0.48, *p* = 0.007) and depression (β = 0.71, *p* < 0.001). Multicollinearity was particularly high in this analysis, and it should therefore be interpreted with caution. The IPA, PAAQ, 6-PAQ, PSPF, and the Cognitive Defusion and Committed Action subscales of the PPF predicted negative parenting practices over and beyond general psychological flexibility and distress, but in a model controlling for one another, only the 6-PAQ remained a significant predictor (β = 0.36, *p* < 0.001). Due to a high level of multicollinearity, this model was also tested without the Cognitive Defusion subscale of the PPF or the PSPF. In this analysis, the IPA (β = 0.24, *p* = 0.041) and the Committed Action subscale of the PPF (β = −0.22, *p* = 0.006) also remained significant predictors in addition to the 6-PAQ. The 6-PAQ and the Acceptance subscale of the PPF significantly predicted positive parenting over and above general psychological flexibility and distress, and both remained significant predictors when controlling for one another (6-PAC β = −0.21, *p* = 0.012; PPF-AC β = 0.60, *p* < 0.001). See Appendix A for additional details on the full analyses.

### 3.2. Study 2 Results: Confirmatory Factor Analysis with the Clinical Sample

After the removal of one outlier, a total of 153 parents of TGD youth presenting to a pediatric gender clinic were included in the CFA and reliability analysis for the IPA. A subset of 124 of these cases in which matched adolescent data were available was used for additional correlation analyses. A large majority of the sample consisted of mothers (over 85%) compared to fathers and nonbinary parents, which is consistent with research in clinical populations. Approximately half of the youth sample identified as male/transmasculine (48.4%) compared to female/transfeminine or nonbinary, which is similarly consistent with research conducted in pediatric gender clinics ([33]). See Table 4 for a full summary of participant characteristics.

#### 3.2.1. Confirmatory Factor Analysis

The IPA was found to be appropriate for factor analysis as evidenced by Bartlett’s Test of Sphericity, *X^2^*(28) = 358.23, *p* < 0.001, and the Kaiser–Meyer–Olkin (KMO) measure, KMO = 0.79. Overall, Model fit was modest (*X^2^*(20) = 68.99, *p* < 0.001; CFI = 0.84; TLI = 0.77, RMSEA (90% CI) = 0.13), with all items except items 6 and 8 demonstrating good model fit with significance levels all ≤0.001 and strong item loading values. Both items (6, 8) had poor factor loadings, with item 6 being the main impact on model fit, with a factor loading value of 0.02 (compared to item 8, which had a factor loading value of 0.24). Modification indices also suggested that correlating errors for items 1 and 2 would result in a significantly better fitting model (i.e., these items may have been responded to in a similar way for reasons other than the content). The model was re-run without item 6 and with the errors for items 1 and 2 correlated, and the resulting model demonstrated a good fit (*X^2^*(13) = 20.57, *p* = 0.082; CFI = 0.97; TLI = 0.96, RMSEA (90% CI) = 0.06). See Table 5 for CFA loadings of the final CFA Model.

To test the post-hoc hypothesis that item 6 may have had population-specific implications, a CFA was run on the Study 1 data with all eight items and found to have adequate fit (*X^2^*(20) = 47.20, *p* = 0.001,; CFI = 0.94; TLI = 0.91, RMSEA (90% CI) = 0.08) with all items demonstrating strong factor loadings (including item 6 with a loading of 0.53).

#### 3.2.2. Reliability and Validity

Internal consistency reliability for the IPA was acceptable for the 8-item version (α = 0.74) and the 7-item version of the measure (α = 0.78). See Table 6 for a summary of variable descriptive statistics and the correlation matrix. While Skewness and Kurtosis were within normal limits for all variables, PAGES-Y scores were more skewed and kurtotic compared to other variables, suggesting that this sample of adolescents viewed their parents as more supportive than may be typical. As expected, adolescent depression and anxiety were significantly positively correlated with parental non-affirming behaviors and significantly negatively correlated with parental acceptance of gender identity. Similarly, parenting inflexibility as measured by the IPA was significantly correlated with both subscales of parental support in the expected directions. While both versions of the IPA were significantly correlated with all variables, slightly stronger correlations were found for the 8-item version of the measure. On an individual item level, two items were significantly correlated with parental support outcomes in the expected directions: “My emotions cause problems in my relationship with my child” and “My worries about my child get in the way of successful parenting”.

## 4. Discussion

The primary goal of the present study was to create a brief tool to assess parenting inflexibility in clinical settings, informed by items from previously established measures. Prior research has demonstrated strong reliability and validity among existing measures, but there are limited indicators for the use of one measure over the other in either research or clinical settings ([6]). The Impact of Parenting Avoidance (IPA) scale was developed using a thematic synthesis of existing measures of parenting flexibility, with a strong focus on ease of administration and interpretation, reading level and accessibility, and clinical utility in a pediatric health setting, such as a multidisciplinary gender clinic. Content validity was established during the measurement development phase using subject matter experts. Additional psychometric properties were evaluated in a general sample of parents as well as a clinical population of parents of TGD youth seeking gender-affirming healthcare.

The first study of psychometric properties was conducted with parents recruited from the general population using MTurk. Among the advantages of this recruitment strategy, this sample represented an equal number of mothers and fathers, consistent with other studies that have used this method ([2]). The sample was also racially and ethnically diverse, but it was somewhat over-representative of parents with a higher socioeconomic status. An exploratory factor analysis with this population indicated that the IPA likely represented a single factor and identified one item as not loading onto the overall construct. After the removal of this item, the IPA demonstrated strong reliability and validity in this sample. When investigating the removed item (“I try hard to avoid having my child feel sad or anxious”), we posited that the item may not have been a good fit based on wording complexity or a general tendency of respondents to rate this item higher than other items (i.e., the mean response was higher with a smaller standard deviation compared to other items).

Our analyses with this general parenting population demonstrated that the IPA had adequate psychometric properties in comparison with other measures of parental flexibility, distress, and parenting practices. When controlling for general psychological flexibility as well as depression, anxiety, and stress, the IPA remained a significant predictor of negative (but not positive) parenting practices. This suggests that it is likely an effective tool for identifying parenting inflexibility that may be playing a role in problematic parenting. This finding is particularly promising as the goal of this measure is to identify targets of intervention for parents of TGD youth seeking medical care. Notably, discriminant validity was not supported in the current study. First, the IPA was unexpectedly positively correlated with social support. However, the measure chosen for this analysis (PROMIS social support) also correlated in unexpected ways with a variety of other measures in the analysis without sound theoretical or practical reasons, and therefore, the measure itself may have been problematic. Second, particularly high correlations among some variables (α > 0.90) further suggest potentially poor discriminant validity ([14]).

We found noteworthy comparisons among the IPA and other measures of parenting flexibility that can be useful to clinicians and researchers when selecting scales for their specific purposes. These results may be clinically useful, but should nonetheless be interpreted with caution due to elevated multicollinearity between the variables. When controlling for other measures of parenting flexibility, the 6-PAQ predicted both positive and negative parenting to a stronger degree compared to the IPA and other flexibility measures, suggesting it may be the best measure for clinicians and researchers who are specifically interested in the way that parenting flexibility affects parenting practices, especially if a longer measure is not contraindicated and/or if examining individual components of parenting flexibility is useful. In a setting where a brief measure aimed at identifying points of intervention (i.e., inflexibility and negative parenting), the IPA is still a suitable measure.

The PSPF was the most successful predictor of anxiety and depression compared to other measures of flexibility. However, this measure was created to mimic the AAQ-II, which has been criticized for potentially assessing distress rather than a fully distinct construct, which may partially explain this finding. The PSPF is similarly brief and easy to administer, so clinicians and researchers may choose this measure when anxiety and depression are important outcomes, keeping the prior caution in mind. The PAAQ was an effective predictor of negative parenting, but not above other measures. The PPF subscales differentially predicted various outcome measures, but neither consistently nor above other measures of parenting flexibility. Both of these measures may be chosen for the specific subscales that are captured when scientifically or clinically indicated, but they are likely less of an effective choice when ease of administration and interpretation is a primary goal.

In the second study, we investigated the psychometric properties of the IPA within a clinical sample of parents of TGD youth presenting to a psychology appointment within a multidisciplinary gender clinic. A confirmatory factor analysis of the 8-item one-factor version of the IPA identified in Study 1 initially resulted in a poorly fitting model, with one item in particular not significantly loading onto the overall factor: “I don’t let my child do things that might upset them or that they might regret”. Many parents of TGD youth seeking medical care present with concerns specifically related to whether or not their child will regret treatment in the future ([3]). Over time, by correcting myths and providing psychoeducation, most of these parents come to realize this is an especially rare outcome ([16]). The sample included in the present study was made up primarily of families early in their care with the clinic, and therefore, regret may be an outsized concern. In other words, parents’ feelings about regret specifically may have influenced their response independent of their overall parenting inflexibility.

In addition to removing this item, we correlated the error of the first two items on the measure based on modification indices. These combined adjustments resulted in a good-fitting model. Correlated error terms among the first two items may be related to a subtheme they both share, but it is more likely that being the first two items on the measure and next to each other resulted in this correlated error. We also post-hoc decided to run the same CFA with the sample of parents in the general population from Study 1 to further explore the role of this item in the overall fit and we found that the model fit well with this first sample, with all items loading onto the IPA factor. This further supports the possibility that Item 6 may be capturing outsized concerns specific to parents of TGD youth.

Within the clinical sample of parents of TGD youth, the reliability of both the 7-item and 8-item versions of the measure was acceptable. Regarding perceived parental support, the sample appeared to be slightly skewed toward more parental support than may be typical, which is consistent with research showing that parents must be at least supportive enough to bring their child to a gender clinic seeking care ([50]). As expected, perceived parental support (both non-affirming and accepting behaviors) was significantly associated with adolescent anxiety and depression. This result adds to the existing literature that parental support and acceptance of their child’s gender identity are important protective factors against negative mental health outcomes. A lack of support, conversely, is a clear clinical target for behavioral health providers working with parents of TGD youth.

Perceived parental support was also predicted by the IPA, suggesting that parenting inflexibility may interfere with a parent’s ability to engage in supportive gender-affirming behaviors toward their child. This correlation was slightly stronger with the 8-item measure. This further suggests that the question related to regret may be more relevant to the TGD youth population and variables of interest, while not fitting well within the construct of parenting inflexibility captured by the IPA.

Together, these two studies support the use of the IPA in identifying parenting inflexibility in clinical settings such as a pediatric gender clinic. The findings further suggest a relationship between inflexibility and lower perceived parental support, a known risk factor for negative mental health outcomes among TGD youth. Clinicians who identify parenting inflexibility in this context can use well-established interventions such as ACT to improve parenting flexibility. Brief ACT interventions have been effective in improving flexibility in parents of children with chronic conditions (e.g., [27]) and with adolescents in pediatric healthcare settings (e.g., [43]). Similar brief interventions may prove useful with parents who are struggling with the acceptance of their TGD child and/or struggling to demonstrate their support. Ultimately, the wellbeing of these parents is also important, and parenting inflexibility may have other consequences (e.g., parental depression or anxiety) that would benefit from intervention.

The measurement development process used in this study was limited by a lack of qualitative data collected from parents. Content validity was established by soliciting feedback from professionals with clinical and research expertise related to psychological flexibility and parenting, but no qualitative data were used in initial item generation. Future research may consider qualitative interviews with parents and clinicians to elicit additional feedback on aspects of parenting inflexibility that may be most relevant, including within a pediatric healthcare setting specifically.

The relevant limitations of the validation studies include sample size, sample representativeness, and differing study methods across the two populations. The MTurk sample was slightly more representative of a higher SES, and the clinical sample was slightly more representative of higher perceived parental support than may be typical in the general population of parents of TGD youth. Parents in the clinical sample also represent one geographic area, and these results may not generalize to parents of TGD youth in other areas, especially those facing heightened stress or stigmatization.

The assessment format was different across the two studies, with the clinical data representing a variety of completion methods (e.g., on paper versus electronically). Sample sizes were relatively low across both studies but sufficient for all analyses that were conducted. Self-report measures are subject to individual bias, as evidenced by parents in Study 2 responding to a particularly relevant item differently than to several other items with highly similar content. Results from factor analyses should be interpreted carefully and in context when determining the strengths and weaknesses of a particular scale with a particular population.

## 5. Conclusions

The Impact of Parenting Avoidance (IPA) scale is a brief measure of parenting inflexibility designed to be easily administered and clinically useful within pediatric healthcare settings (see Appendix A for the final version of the IPA). Exploratory and confirmatory factor analyses were used across two studies to validate an optimized 7-item measure. Reliability and validity were demonstrated in both studies, and the IPA was able to predict relevant parent variables (Study 1) and child variables (Study 2). Clinicians may find this measure useful in efficiently assessing parenting inflexibility to identify parents who may benefit from acceptance and mindfulness-based interventions. This is the first study to examine parenting inflexibility in parents of TGD youth, which may be an important target of intervention for improving not only parent wellbeing, but also subsequent support and acceptance of their youth, a significant protective factor in the wellbeing of these youth.

## Figures and Tables

**Table 1 behavsci-15-00625-t001:** Characteristics of existing measures of parenting flexibility.

	Items	Response Format	Total Score	Subscales	Reverse Scoring
PAAQ	15	(1) Never true to (7) Always true	Parenting Inflexibility	Unwillingness; Inaction	Some items are reverse scored
PPF	19	(1) Never true to (7) Always true	N/A	Acceptance; Cognitive defusion; Committed action	Some items are reverse scored
6-PAQ	18	(1) Strongly disagree/never to (4) Strongly agree/almost always	Parenting Inflexibility	Acceptance; Defusion; Being present; Self as context; Values; Committed Action	Some items are reversed scored
PSPF	7	(1) Never true to (7) Always true	Parenting Flexibility	N/A	All items are reverse scored

Note. PAAQ = Parental Acceptance and Action Questionnaire; PPF = Parental Psychological Flexibility Questionnaire; 6-PAQ = Parental Acceptance Questionnaire; PSPF = Parenting Specific Psychological Flexibility Scale.

**Table 2 behavsci-15-00625-t002:** Items included in validation analyses with individual reading level scores.

Impact of Parenting Avoidance Item	Flesch-Kincaid
1. My painful thoughts and feelings get in the way of how I want to parent.	4.42
2. My emotions cause problems in my relationship with my child.	7.19
3. I dwell on my parenting and what I would do differently next time.	5.82
4. It’s hard to make parenting decisions because I’m afraid of making a mistake.	9.14
5. My worries about my child get in the way of successful parenting.	6.79
6. I don’t let my child do things that might upset them or they might regret.	3.63
7. I worry about how other people might judge my parenting.	7.19
8. Before I allow my child to do something, I must have all my fears worked out.	6.14
9. I try hard to avoid having my child feel sad or anxious.	3.84

**Table 3 behavsci-15-00625-t003:** Study 1 sample characteristics.

Age, Range (Mean)		25–59 (35.7)
Gender ^a^		
	Female	51
	Male	49
Racial/Ethnic Identity		
	American Indian or Alaska Native	1
	Asian	10
	Black or African American	8
	Hispanic, Latino, or Spanish origin	25
	Middle Eastern or North African	0
	Native Hawaiian or Other Pacific Islander	2
	White	52
	Multiracial	3
Relationship status		
	Single (never married, divorced, separated, widowed)	9
	Married or living with partner	92
Highest level of education		
	Less than high school graduate	1
	High school graduate, diploma, or the equivalent (e.g., GED)	5
	Some college or associate degree	15
	Bachelor’s degree	54
	Master’s, professional, or doctorate degree	26

^a^ Gender was not reported for one participant.

**Table 4 behavsci-15-00625-t004:** Study 2 sample characteristics.

		Parent			Child		
		N	Gender		Age Range (*M*)	Gender	
Full Sample		153	n	%	8–18 (14.3)	n	%
	Male		14	9.2		74	48.4
	Female		134	87.6		46	30.1
	Nonbinary		5	3.3		33	21.6
Matched Subset		124			11–17 (14.6)		
	Male		13	10.5		66	48.4
	Female		106	85.5		37	29.8
	Nonbinary		5	4.0		27	21.0

**Table 5 behavsci-15-00625-t005:** Impact of Parenting Avoidance (IPA) confirmatory factor loadings.

Item	Estimate	Std. Error	*p*-Value
IPA 1	0.59	0.07	<0.001
IPA 2	0.63	0.06	<0.001
IPA 3	0.66	0.07	<0.001
IPA 4	0.61	0.07	<0.001
IPA 5	0.75	0.07	<0.001
IPA 7	0.45	0.08	<0.001
IPA 8	0.34	0.07	<0.001

**Table 6 behavsci-15-00625-t006:** Study 2: descriptive statistics and correlation matrix.

	*N*	Range	*M* (*SD*)	1	2	3	4
1. PHQ-9	121	0–27	10.6 (7.1)	-	-	-	-
2. GAD-7	119	0–21	8.5 (5.9)	0.76 **	-	-	-
3. PAGES-Y NA	118	8–32	12.3 (5.8)	0.30 **	0.24 *	-	-
4. PAGES-Y AC	117	9–30	24.1 (4.9)	−0.32 **	−0.20 *	−0.80 **	-
5. IPA-8	153	8–31	17.3 (4.2)	0.22 *	0.08	0.22 *	−0.26 **
6. IPA-7	153	7–28	14.9 (4.0)	0.21 *	0.07	0.20 *	−0.26 **

* *p* < 0.05. ** *p* < 0.01.

## Data Availability

The data presented in this study are available on request from the corresponding author due to institutional requirements.

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
