# Peer review of "The Impact of Parenting Avoidance (IPA): Scale Development and Psychometric Evaluation Among Parents of Transgender Youth"

_behavsci, 2025, doi:10.3390/bs15050625_

Round 1
Reviewer 1 Report
Comments and Suggestions for Authors
I appreciate the opportunity to review your manuscript. The development of a new measure surrounding parenting inflexibility and the impacts for TGD youth. I have provided some feedback below for you to consider.
Towards the end of the introduction in the justification for needing a new measure, there is a statement about needing a new measure to " identify clinically relevant parenting inflexibility efficiently". I am unclear about this and would appreciate more definition and description around what is meant by clinically relevant and/or parenting inflexibility. Does this refer to clinic aspects of parenting? or aspects of parenting that may be impactful for clinical relevance.
The manuscript describes a scoping review and other methods to arrive at a series of items for the new measure; however, there is no mention of any qualitative assessment. Was a qualitative component considered as part of this study? If other measures exist and you pull from them, as opposed to ask clinicians and parents about experiences, is it really developing a new measure? Or is it just a modified version of the priors?
In the discussion/conclusion it would be beneficial to discuss the impacts of this new measure for TGD youth. Specifically, it would be beneficial to tie the information back to how this new measure would improve the clinical outcomes for TGD youth, or more specifically address the direct implications for these individuals.
Author Response
Response to Reviewer 1 Comments
Thank you very much for taking the time to review our manuscript, entitled “The Impact of Parenting Avoidance (IPA): Scale Development and Psychometric Evaluation among Parents of Transgender Youth.” Please find the detailed responses below and the corresponding revisions marked in red in the re-submitted files.
Point-by-point response to Comments and Suggestions for Authors
I appreciate the opportunity to review your manuscript. The development of a new measure surrounding parenting inflexibility and the impacts for TGD youth. I have provided some feedback below for you to consider.
Comment 1: Towards the end of the introduction in the justification for needing a new measure, there is a statement about needing a new measure to " identify clinically relevant parenting inflexibility efficiently". I am unclear about this and would appreciate more definition and description around what is meant by clinically relevant and/or parenting inflexibility. Does this refer to clinic aspects of parenting? or aspects of parenting that may be impactful for clinical relevance.
Response 1: Thank you for pointing out that this section could use clarification. We agree and have revised the introduction to more explicitly define what is clinically useful about identifying parenting inflexibility. These revisions can be found on page 3, lines 117-125.
Comment 2: The manuscript describes a scoping review and other methods to arrive at a series of items for the new measure; however, there is no mention of any qualitative assessment. Was a qualitative component considered as part of this study? If other measures exist and you pull from them, as opposed to ask clinicians and parents about experiences, is it really developing a new measure? Or is it just a modified version of the priors?
Response 2: Because items were revised and additional items were added to specifically capture the two identified aspects of (1) avoidance and (2) impact, we do consider this to be novel measure rather than a modified version of an existing measure. However, we do agree that the lack of qualitative data is a limitation. While we involved experts in the field in the content validation and item revision phase, we did not collect qualitative data from experts or parents. We have added this limitation to our discussion, and this new paragraph can be found on page 16, starting on lines 648-654.
Comment 3: In the discussion/conclusion it would be beneficial to discuss the impacts of this new measure for TGD youth. Specifically, it would be beneficial to tie the information back to how this new measure would improve the clinical outcomes for TGD youth, or more specifically address the direct implications for these individuals.
Response 3: Thank you so much for this comment. We have added to the discussion in attempt to highlight the clinical purpose of the study, which is supporting the wellbeing of transgender youth and their families. These revisions can be found on page 14, lines 569-570, page 15, lines 625-632, and the paragraph starting at line 636.
Reviewer 2 Report
Comments and Suggestions for Authors
Thank you for your paper. In my view, the paper is vague. The English language used in this paper is unclear.
Several examples:
Abstract: "Parental support and acceptance are strong protective factors for transgender
and gender diverse youth.". Protective factors for what? The sentence in unclear.
"Psychological inflexibility specifically in the role of parenting, or “parenting inflexibility,” may be a useful target of intervention for this population". "For this population" means for transgender
and gender diverse youth. The sentence in unclear. The authors seem to refer to parents of transgender and gender diverse youth here.
"Study 1 was an exploratory factor analysis with parents in the general population recruited using MTurk". A study is an analysis? Factor analysis with parents? Factor analysis usually use items, statements, but not parents.
"Study 2 was a confirmatory factor analysis with parents of transgender youth recruited from a clinic". The same problem.
"The resulting measure demonstrated moderately high reliability, and it was significantly correlated with both parent and adolescent outcomes. ". Moderately high? Sounds awkward. What does this mean? "both parent and adolescent outcomes." What does this mean?
The authors mentioned "parenting inflexibility" many times in the abstract and in the paper, but the title of the scale refers to "Impact of Parenting Avoidance (IPA)". It is unclear why the terms are used in such a mixed way.
Dear Authors,
In my view, in the current form, due to a high number of the language issues, the paper cannot be reviewed. It is unclear what the authors would like to express in their study.
Comments on the Quality of English Language
Please see the review.
Author Response
Response to Reviewer 2 Comments
Thank you very much for taking the time to review the abstract of our manuscript, entitled “The Impact of Parenting Avoidance (IPA): Scale Development and Psychometric Evaluation among Parents of Transgender Youth.” Please find the detailed responses below and the corresponding revisions marked in red in the re-submitted files.
Point-by-point response to Comments and Suggestions for Authors
Comments 1: Thank you for your paper. In my view, the paper is vague. The English language used in this paper is unclear.
Several examples:
Abstract: "Parental support and acceptance are strong protective factors for transgender and gender diverse youth.". Protective factors for what? The sentence in unclear.
"Psychological inflexibility specifically in the role of parenting, or “parenting inflexibility,” may be a useful target of intervention for this population". "For this population" means for transgender and gender diverse youth. The sentence in unclear. The authors seem to refer to parents of transgender and gender diverse youth here.
"Study 1 was an exploratory factor analysis with parents in the general population recruited using MTurk". A study is an analysis? Factor analysis with parents? Factor analysis usually use items, statements, but not parents.
"Study 2 was a confirmatory factor analysis with parents of transgender youth recruited from a clinic". The same problem.
"The resulting measure demonstrated moderately high reliability, and it was significantly correlated with both parent and adolescent outcomes. ". Moderately high? Sounds awkward. What does this mean? "both parent and adolescent outcomes." What does this mean?
The authors mentioned "parenting inflexibility" many times in the abstract and in the paper, but the title of the scale refers to "Impact of Parenting Avoidance (IPA)". It is unclear why the terms are used in such a mixed way.
Dear Authors,
In my view, in the current form, due to a high number of the language issues, the paper cannot be reviewed. It is unclear what the authors would like to express in their study.
Response 1: Thank you again for reviewing our abstract. We understand that in an attempt to keep the abstract succinct, we left several pieces of information implied and not explicitly stated. We have revised the entire abstract (page 1) to address your feedback above. We also went through the manuscript and revised content that reflected the problems you identified in the abstract.
With regard to your specific comment about “moderately high” reliability, we agree that this was an imprecise term. Rather than using descriptors such as high, moderate, and low, we have chosen to revise the entire manuscript to reflect the more commonly used reliability descriptors of excellent, good, acceptable, questionable, and poor. We appreciate the opportunity to improve the manuscript with this revision.
With regard to your specific comment about the title of the measure, the abstract has been revised to specify that parenting inflexibility is characterized by avoidance. However, the full reason for measure title would cause the abstract to exceed its word limit. As you will see in the text, the results of our measurement synthesis suggested that two common themes of items were that they included an aspect of avoidance and the impact of that avoidance. The items in the measure were therefore written to specifically capture those two broad aspects, hence the measure name.
In addition to the specific clarifications requested in the abstract, we have reviewed the entire manuscript and made small revisions throughout to improve readability. We are happy to make any additional clarifications to language that may be requested.
Round 2
Reviewer 1 Report
Comments and Suggestions for Authors
Thank you for your thoughtful revisions to the manuscript and for addressing the suggestions I had raised previously. I appreciate your responses and note the revisions do address these comments well. I do not have any further concerns at this time.
Author Response
Response to Reviewer 1 Comments – Round 2
Thank you very much again for taking the time to review our manuscript, entitled “The Impact of Parenting Avoidance (IPA): Scale Development and Psychometric Evaluation among Parents of Transgender Youth.”
Reviewer 2 Report
Comments and Suggestions for Authors
1. Thank you fro amending English.
2. "Regression analysis found strong convergent validity with measures of psychological inflexibility, including the PAAQ (α = .79, p < .001), 6-PAQ (α = .82, p < .001), and PSPF (α = .90, p <.001). The IPA was also negatively correlated with psychological flexibility on the Cognitive Defusion (α = -.91, p < .001) and Committed Action (α = -.88, p < .001) subscales of the
PPF, but not the Acceptance subscale (α = -.04, p = .681). Contrary to our hypothesis, the
IPA was correlated with social support (α = .69, p < .001), and we were unable to determine
discriminate validity in this study. Social support was also correlated in unexpected directions with other measures of distress (DASS α = .71, p < .001) and psychological flexibility (AAQ α = .76, p < .001; PAA".
Please could you explain what does "a" mean here? If you are writing about regression coefficients, this must be "β" (beta).
3. Moreover, correlations with other variables are very strong, suggesting issues with discriminant validity. Please elaborate on this
Please see: Cheung, G.W., Cooper-Thomas, H.D., Lau, R.S. et al. Reporting reliability, convergent and discriminant validity with structural equation modeling: A review and best-practice recommendations. Asia Pac J Manag 41, 745–783 (2024). https://doi.org/10.1007/s10490-023-09871-y
4. Lines 465-473: The same problem. In general, regression analysis should be presented in a common way with B, beta, t, p, F with df. It can be done in Supplementary Materials.
5. Lines 474-486: Please examine linear regression assumptions here and above in the paper (normality, absence of collinearity etc.).
6. Please add scoring instructions for the measure (in Supplementary Materials).
Author Response
Response to Reviewer 2 Comments – Round 2
Thank you very much again for taking the time to review the abstract of our manuscript, entitled “The Impact of Parenting Avoidance (IPA): Scale Development and Psychometric Evaluation among Parents of Transgender Youth.” Please find the detailed responses below and the corresponding revisions marked in red in the re-submitted files.
Point-by-point response to Comments and Suggestions for Authors
Comment 1. Thank you for amending English.
Response 1. Thank you for your initial feedback and helping to improve the readability.
Comment 2. "Regression analysis found strong convergent validity with measures of psychological inflexibility, including the PAAQ (α = .79, p < .001), 6-PAQ (α = .82, p < .001), and PSPF (α = .90, p <.001). The IPA was also negatively correlated with psychological flexibility on the Cognitive Defusion (α = -.91, p < .001) and Committed Action (α = -.88, p < .001) subscales of the
PPF, but not the Acceptance subscale (α = -.04, p = .681). Contrary to our hypothesis, the
IPA was correlated with social support (α = .69, p < .001), and we were unable to determine
discriminate validity in this study. Social support was also correlated in unexpected directions with other measures of distress (DASS α = .71, p < .001) and psychological flexibility (AAQ α = .76, p < .001; PAA".
Please could you explain what does "a" mean here? If you are writing about regression coefficients, this must be "β" (beta).
Response 2. Thank you for pointing out this error. This was meant to say correlation analysis. Page 11, Line 454 has been revised to reflect this correction. A sentence was also added to lines 468-469 to specify the shift from our correlation analyses to our regression analyses. We also found a few other places in the manuscript that included this error, which we have revised, including: Page 11, Line 465; Page 8, Lines 359 and 361; Page 10, Line 427; Page 12, Line 492.
Comment 3. Moreover, correlations with other variables are very strong, suggesting issues with discriminant validity. Please elaborate on this
Please see: Cheung, G.W., Cooper-Thomas, H.D., Lau, R.S. et al. Reporting reliability, convergent and discriminant validity with structural equation modeling: A review and best-practice recommendations. Asia Pac J Manag 41, 745–783 (2024). https://doi.org/10.1007/s10490-023-09871-y
Response 3. We appreciate you bring this us and pointing us to an appropriate reference. Pages 14-15, Lines 587-594 of the discussion have been revised to include this additional issue with discriminant validity.
Comment 4. Lines 465-473: The same problem. In general, regression analysis should be presented in a common way with B, beta, t, p, F with df. It can be done in Supplementary Materials.
Comment 4. Thank you again. As described above in Response 2, the error was in reporting the analysis, which we have revised.
Comment 5. Lines 474-486: Please examine linear regression assumptions here and above in the paper (normality, absence of collinearity etc.).
Comment 5. We appreciate this suggestion. In addition to reporting on assumptions, we realized that one aspect of multicollinearity could be addressed by combining distress scores, which were not necessary to keep as separate variables (compared to our primary measures of interest, which were inherently related but necessary to examine as separate variables). We have reported on assumptions, made this change and updated relevant statistical results, and added reminders to the results and discussion to interpret results in the context of expected collinearity. These revisions can be found on Pages 11-12, Lines 469-501 and Page 15, Lines 597-598.
Comment 6. Please add scoring instructions for the measure (in Supplementary Materials).
Response 6. We have make the scoring instructions more clearly stated in the supplemental material along with the measure.
Round 3
Reviewer 2 Report
Comments and Suggestions for Authors
Thank you for the corrections. In my view, there are still errors which should preclude the publication of this paper.
- I still do not understand why the authors use an incorrect letter alpha (α) to indicate the correlation coefficients. I would like to kindly ask the authors to address these errors adequately for the second time (lines 454-464 and above in the paper). The authors should use a letter r to indicate correlation coefficients.
2. Moreover, all values (except p-values) should be reported with two decimal places. P-values should be reported with three decimal places. Sometimes the authors reported three decimal places, sometimes only one, sometimes two.
3. Please report regression analysis in a form of a table, with B, beta, t, p, F-values, and R-squared. I am asking for this twice. The authors declined to edit their paper according this comment, without justification. Please present analyses according to common standards in the field. The is no space limit, as such these analyses should be presented in the paper or in Supplementary Materials.
Author Response
Response to Reviewer 2 Comments – Round 3
Thank you very much again for taking the time to review the abstract of our manuscript, entitled “The Impact of Parenting Avoidance (IPA): Scale Development and Psychometric Evaluation among Parents of Transgender Youth.” Please find the detailed responses below and the corresponding revisions marked in red in the re-submitted files.
Point-by-point response to Comments and Suggestions for Authors
Thank you for the corrections. In my view, there are still errors which should preclude the publication of this paper.
Comment 1. I still do not understand why the authors use an incorrect letter alpha (α) to indicate the correlation coefficients. I would like to kindly ask the authors to address these errors adequately for the second time (lines 454-464 and above in the paper). The authors should use a letter r to indicate correlation coefficients.
Response 1. Yes, thank you for recognizing that this was still an error. We have revised the correlation results to use the letter r. Page 11, Lines 456-468
Comment 2. Moreover, all values (except p-values) should be reported with two decimal places. P-values should be reported with three decimal places. Sometimes the authors reported three decimal places, sometimes only one, sometimes two.
Response 2. Thank you for finding these inconsistencies. With the exception of p-values, all inferential statistics were revised to 2 decimal places. Specifically, Page 11, Lines 474-477, Page 12, Lines 493-505, Page 13, Lines 521-539 and Table 5, and Page 14, Table 6. Percentage and mean values for data measured on an integer scale (e.g., age) were kept at one decimal place for appropriate statistical precision.
Comment 3. Please report regression analysis in a form of a table, with B, beta, t, p, F-values, and R-squared. I am asking for this twice. The authors declined to edit their paper according this comment, without justification. Please present analyses according to common standards in the field. The is no space limit, as such these analyses should be presented in the paper or in Supplementary Materials.
Response 3. We apologize for missing this comment in the prior feedback. Given the number of analyses we ran, we have chosen to present these tables in the supplemental materials. We have created supplementary tables S2-S9 to include this information and alluded to the tables within the manuscript on Page 11, Lines 477-478, and Page 12, Lines 505-506.
Round 4
Reviewer 2 Report
Comments and Suggestions for Authors
The authors have edited the paper according to reviewer's recommendations. In my view, it seems to be clear. For future publications, I would suggest carefully checking the journal's requirements. For instance, the use of zeroes before full stops in numbers (for example, correct 0.05, not .05). Hope the authors amend this issue during the final edits of their paper (if the paper is accepted).
Author Response
Response to Reviewer 2 Comments – Round 4
Reviewer Comment: The authors have edited the paper according to reviewer's recommendations. In my view, it seems to be clear. For future publications, I would suggest carefully checking the journal's requirements. For instance, the use of zeroes before full stops in numbers (for example, correct 0.05, not .05). Hope the authors amend this issue during the final edits of their paper (if the paper is accepted).
Response: Thank you very much again for taking the time to review the abstract of our manuscript, entitled “The Impact of Parenting Avoidance (IPA): Scale Development and Psychometric Evaluation among Parents of Transgender Youth.” We appreciate you pointing out this instruction which was missed. We have revised the text, tables, and supplementary tables to include leading zeroes before all decimal numbers.